# Development of a Method for the Fast Detection of Extended-Spectrum β-Lactamase- and Plasmid-Mediated AmpC β-Lactamase-Producing *Escherichia coli* and *Klebsiella pneumoniae* from Dogs and Cats in the USA

**DOI:** 10.3390/ani13040649

**Published:** 2023-02-13

**Authors:** Kwang-Won Seo

**Affiliations:** College of Veterinary Medicine, Chungbuk National University, Cheongju 28644, Republic of Korea; vetskw16@cbnu.ac.kr

**Keywords:** *Escherichia coli*, *Klebsiella pneumoniae*, antimicrobial resistance, clinic, extended-spectrum β-lactamase, plasmid-mediated AmpC β-lactamase

## Abstract

**Simple Summary:**

Extended-spectrum β-lactamase (ESBL) and plasmid-mediated AmpC (pAmpC) β-lactamase-producing *Escherichia coli* and *Klebsiella pneumoniae* cause treatment failures in veterinary medicine. Many methods have been recommended for the detection of ESBL and pAmpC β-lactamase production but they are very subjective and the appropriate facilities are not available in most laboratories, especially not in clinics. We report the development of a method that can detect ESBL- and pAmpC β-lactamase-producing bacteria and this method is a fast, and low-cost tool for the screening of frequently encountered ESBL- and pAmpC β-lactamase-producing bacteria and would assist in diagnosis and improve therapeutic treatment in animal hospitals.

**Abstract:**

Antibiotic resistance, such as resistance to beta-lactams and the development of resistance mechanisms, is associated with multifactorial phenomena and not only with the use of third-generation cephalosporins. Many methods have been recommended for the detection of ESBL and pAmpC β-lactamase production but they are very subjective and the appropriate facilities are not available in most laboratories, especially not in clinics. Therefore, for fast clinical antimicrobial selection, we need to rapidly detect ESBL- and pAmpC β-lactamase-producing bacteria using a simple method with samples containing large amounts of bacteria. For the detection of ESBL- and pAmpC phenotypes and genes, the disk diffusion test, DDST and multiplex PCR were conducted. Of the 109 samples, 99 (90.8%) samples were grown in MacConkey broth containing cephalothin, and 71 samples were grown on MacConkey agar containing ceftiofur. Of the 71 samples grown on MacConkey agar containing ceftiofur, 58 *Escherichia coli* and 19 *Klebsiella pneumoniae* isolates, in particular, harbored β-lactamase genes. Of the 38 samples that did not grow in MacConkey broth containing cephalothin or on MacConkey agar containing ceftiofur, 32 isolates were identified as *E. coli*, and 10 isolates were identified as *K. pneumoniae*; β-lactamase genes were not detected in these *E. coli* and *K. pneumoniae* isolates. Of the 78 ESBL- and pAmpC β-lactamase-producing *E. coli* and *K. pneumoniae*, 55 (70.5%) isolates carried one or more ESBL genes and 56 (71.8%) isolates carried one or more pAmpC β-lactamase genes. Our method is a fast, and low-cost tool for the screening of frequently encountered ESBL- and pAmpC β-lactamase-producing bacteria and it would assist in diagnosis and improve therapeutic treatment in animal hospitals.

## 1. Introduction

*Escherichia coli* and *Klebsiella pneumoniae* are members of the Enterobacteriaceae family, which mostly act as commensals in the intestinal tract of animals and humans. In particular, these bacteria can cause community-onset infections in animals and humans and are the common bacteria associated with urinary tract infections [1,2,3,4,5,6]. β-lactams are preferred for treating these infections in humans and veterinary medicine [7,8].

Extended-spectrum β-lactamase (ESBL) and plasmid-mediated AmpC (pAmpC) β-lactamases are plasmid-encoded enzymes which are capable of inactivating a large number of β-lactam antibiotics, including extended-spectrum and very-broad-spectrum cephalosporins [9]. The most common bacteria that carry ESBL and pAmpC β-lactamase genes include *E. coli* and *K. pneumoniae* [10]. The emergence of ESBL- and pAmpC β-lactamase-producing *E. coli* and *K. pneumoniae* in healthy and diseased animals constitutes an increasing challenge to infection management in veterinary therapy [11]. Moreover, the resistance caused by ESBL and pAmpC β-lactamases is usually multidrug resistance, which leads to critical therapeutic limitations [12,13].

Many tests have been recommended for the detection of ESBL and pAmpC β-lactamase production according to phenotype [14,15,16]. The most commonly used methods include the double disc synergy test (DDST) and Clinical and Laboratory Standards Institute (CLSI) confirmatory test. However, these methods are very subjective and can only be used for a single colony isolated from a sample. In addition, molecular methods are key tools in detection; however, the appropriate facilities are not available in most laboratories, especially not in clinics, and they are used for a single colony. Therefore, for fast clinical antimicrobial selection, we need to rapidly detect ESBL- and pAmpC β-lactamase-producing bacteria using a simple method with samples containing large amounts of bacteria. The objective of this study was to develop and evaluate a new detection method for ESBL- and pAmpC β-lactamase-producing bacteria and to evaluate the isolation and characterization of ESBL- and pAmpC β-lactamase-producing *E. coli* and *K. pneumoniae* from shelter dogs and cats using this method.

## 2. Materials and Methods

### 2.1. Study Design 

For the development of an isolation method for ESBL- and pAmpC β-lactamase-producing *E. coli* and *K. pneumoniae*, we used a sample, isolated in 2014, from an animal hospital in Mississippi State University. For the detection of ESBL- and pAmpC β-lactamase-producing bacteria by phenotype, ESBL production was confirmed using the DDST by using a disc of amoxicillin-clavulanate (AMC, 20/10 μg) along with four cephalosporins; cefotaxime (30 μg), ceftriaxone (30 μg), cefpodoxime (10 μg) and cefepime (50 μg), and pAmpC β-lactamase production was evaluated using cefoxitin (30 μg) as an inhibitor of pAmpC enzymes [10]. In addition, multiplex PCR of ESBL and pAmpC β-lactamase genes and the disk diffusion test were conducted to identify β-lactamases genes and cephalosporin resistance patterns [17]. Overall, 26 *E. coli* (each 13 ESBL- and pAmpC β-lactamase-producing bacteria or not) and 46 *K. pneumoniae* (each 23 ESBL- and pAmpC β-lactamase-producing bacteria or not) samples were isolated (Table 1).

Based on information about the use of cephalosporin in animal hospitals, we chose cephalosporin antimicrobials (1 first-generation cephalosporin and 2 third-generation cephalosporins). A total of 26 *E. coli* and 46 *K. pneumoniae* isolates were inoculated into tryptic soy broth (Sigma, St. Louis, MO, USA), and these inoculated cultures were incubated at 37 °C for 4 h. The pre-enriched TSB with bacteria was inoculated into MacConkey broth (Sigma) containing first-generation cephalosporin (cephalothin (128 µg/mL; Sigma)) and incubated at 37 °C for 24 h. After enrichment, only growth-positive MacConkey broth containing cephalothin was streaked on MacConkey agar (Sigma) containing different concentrations of each third-generation cephalosporin (ceftiofur (16, 32, 64 µg/mL; Sigma) and ceftriaxone (16, 32, 64 µg/mL; Sigma)). The results of growth under different concentrations of each cephalosporin are shown in Table 2.

Based on Table 2, we determined the type and concentration of cephalosporins (cephalothin (128 µg/mL) and ceftiofur (32 µg/mL)) for the isolation of ESBL- and pAmpC β-lactamase-producing *E. coli* and *K. pneumoniae*.

### 2.2. Sampling

We sampled dogs and cats in 6 animal shelters in Mississippi between May and August 2019. Dogs and cats eligible for sample collection were those that appeared healthy and caged individually. The feces, oral, and ear samples were collected using sterile cotton swabs and maintained at approximately 4 °C during transport to the research laboratory for processing. 

### 2.3. Isolation of ESBL- and pAmpC β-Lactamase-Producing E. coli and K. pneumoniae

All samples were analyzed following a specific process (Figure 1). After streaking on MacConkey agar plates, we selected the colony which appeared *E. coli* or *K. pneumoniae*. To identify *E. coli* and *K. pneumoniae*, PCR was carried out as previously described [18,19]. For the isolation of ESBL- and pAmpC β-lactamase-producing *E. coli* and *K. pneumoniae*, all *E. coli* and *K. pneumoniae* samples were analyzed by multiplex PCR of ESBL and pAmpC β-lactamase genes as described above.

### 2.4. Antimicrobial Susceptibility Testing

All ESBL- and pAmpC β-lactamase-producing *E. coli* and *K. pneumoniae* isolates were investigated for their antimicrobial resistance with the disc diffusion test using the following discs (BD): AMC (20/10 μg), ampicillin (AM, 10 μg), cefoxitin (30 μg), cefpodoxime (10 μg), chloramphenicol (30 μg), colistin (CT, 10 μg), enrofloxacin (5 μg), gentamicin (G, 10 μg), imipenem (IPM, 10 μg), nalidixic acid (NA, 30 μg), tetracycline (30 μg), and trimethoprim-sulfamethoxazole (1.25/23.75 μg). The results were interpreted according to the CLSI guidelines [20]. *E. coli* ATCC 25922 was used as a control organism in the antimicrobial susceptibility tests. 

## 3. Results and Discussion

### 3.1. Isolation of ESBL and pAmpC-producing Escherichia coli and Klebsiella pneumoniae

A total of 109 samples were analyzed in this study: 77 samples from dogs and 32 samples from cats in 6 shelters in Mississippi. Among the 109 samples, 99 (90.8%) samples were grown in MacConkey broth containing cephalothin, and 71 samples were grown on MacConkey agar containing ceftiofur (Table 3). To identify ESBL- and pAmpC β-lactamase-producing *E. coli* and *K. pneumoniae*, we conducted multiplex PCR of ESBL and pAmpC β-lactamase genes in samples that grew or did not grow on MacConkey agar containing ceftiofur. Of the 71 samples grown on MacConkey agar containing ceftiofur, 58 isolates were identified as *E. coli*, and 20 isolates were identified as *K. pneumoniae*. All *E. coli* isolates harbored β-lactamase genes, and 19 *K. pneumoniae* isolates harbored β-lactamase genes. Among the 38 samples that did not grow in MacConkey broth containing cephalothin or on MacConkey agar containing ceftiofur, 32 isolates were identified as *E. coli*, and 10 isolates were identified as *K. pneumoniae*; β-lactamase genes were not detected in these *E. coli* and *K. pneumoniae* isolates (Table 4). Only one *K. pneumoniae* isolate, which was isolated from MacConkey agar containing ceftiofur, did not carry ESBL and pAmpC β-lactamase genes. Dallenne et al. and Pimenta et al. reported that the absence of ESBL and pAmpC β-lactamase genes may be explained by the presence of a new enzyme due to the high rate of mutations of β-lactamase genes [17,21]. By using this method, we successfully isolated ESBL- and pAmpC β-lactamase-producing *E. coli* and *K. pneumoniae* from all the samples from dogs and cats in just 3 days.

### 3.2. Characterization of ESBL and pAmpC-Producing Escherichia coli and Klebsiella pneumoniae

Of the 78 ESBL- and pAmpC β-lactamase-producing *E. coli* and *K. pneumoniae*, 55 (70.5%) isolates carried 1 or more ESBL genes: CTX-M-1 (24 *E. coli* isolates), CTX-M-2 (4 *E. coli* isolates and 2 *K. pneumoniae* isolates), CTX-M-9 (4 *E. coli* isolates and 3 *K. pneumoniae* isolates), TEM (25 *E. coli* isolates and 4 *K. pneumoniae* isolates), and OXA-1 (7 *E. coli* isolates). In addition, of the 78 ESBL- and pAmpC β-lactamase-producing *E. coli* and *K. pneumoniae* isolates, 56 (71.8%) isolates carried 1 or more pAmpC β-lactamase genes: CIT (43 *E. coli* isolates and 7 *K. pneumoniae* isolates), EBC (4 *E. coli* isolates and 1 *K. pneumoniae* isolate), ACC (1 *E. coli* isolate and 3 *K. pneumoniae* isolates), FOX (1 *E. coli* isolate and 3 *K. pneumoniae* isolates), and DHA (1 *E. coli* isolate) (Table 5).

The major ESBL genes were the CTX-M type, and the major pAmpC β-lactamase genes were the CIT type. These findings are consistent with those of a previous study showing that the occurrence of the CTX-M-type and CIT-type genes was the highest in animals in various countries [22,23,24,25]. In Korea, the distribution of CTX-M type genes in *E. coli* and *K. pneumoniae* isolated from dogs and cats has been reported, and pAmpC β-lactamase genes, especially the CIT type, have also been detected [26]. In Europe, CTX-M type genes as well as plasmid-mediated CIT type genes have been detected in *K. pneumoniae* and *E. coli* from healthy and sick animals including food-producing animals [27,28,29,30,31,32,33,34]. These results indicated that CTX-M- and CIT-type genes have been disseminated throughout the dog and cat populations in the USA.

The other ESBL genes conferring the β-lactam resistance detected in 52.7 and 12.7% of isolates in this study were the TEM and OXA genes, respectively. The TEM and OXA genes were previously identified in clinical *E. coli* and *K. pneumoniae* isolates from companion animals in Europe, which have been found to possess resistance genes against β-lactamase inhibitors (e.g., IRT genes), making such species more of a threat [2,3,32,35,36,37,38].

We also detected various pAmpC β-lactamase genes such as EBC, ACC, FOX, and DHA. These enzymes are pAmpC β-lactamases developed through the transfer of chromosomal genes for inducible AmpC β-lactamases onto plasmids and confer a resistance pattern to most β-lactam antibiotics. Recent reports on *E. coli* and *K. pneumoniae* isolates have shown the prevalence of EBC, ACC, FOX, and DHA-like pAmpC β-lactamases in both human and animal hospitals. The existence of pAmpC β-lactamase genes poses a great challenge to infection control because they can be expressed in larger amounts and have high transmissibility to other bacterial species [39].

### 3.3. Antimicrobial Resistance Phenotypes

Among the 58 ESBL- and pAmpC β-lactamase-producing *E. coli* isolates, the rates of resistance to various antimicrobials were as follows: cefpodoxime (58/58, 100.0%), ceftazidime (58/58, 100.0%), cefotaxime (58/58, 100.0%), AM (57/58, 98.3%), AMC (51/58, 87.9%), trimethoprim-sulfamethoxazole (48/58, 82.8%), CT (47/58, 81.0%), tetracycline (45/58, 77.6%), NA (38/58, 65.5%), cefoxitin (36/58, 62.1%), enrofloxacin (24/58, 41.4%), chloramphenicol (22/58, 37.9%), G (11/58, 19.0%), and IMP (0/58, 0.0%) (Figure 2). In addition, the rates of antimicrobial resistance of 20 *K. pneumoniae* isolates were as follows: cefpodoxime (20/20, 100.0%), ceftazidime (20/20, 100.0%), cefotaxime (20/20, 100.0%), cefoxitin (20/20, 100.0%), AM (20/20, 100.0%), AMC (20/20, 100.0%), CT (19/20, 95.0%), NA (12/20, 60.0%), chloramphenicol (1/20, 5.0%), IMP (0/20, 0.0%), tetracycline (0/20, 0.0%), trimethoprim-sulfamethoxazole (0/20, 0.0%), enrofloxacin (0/20, 0.0%), and G (0/20, 0.0%) (Figure 3). In particular, the rate of resistance to cephalosporins, AM, AMC, CT, and NA was more than 60% for both *E. coli* and *K. pneumoniae*. In addition to the resistance to most β-lactam antibiotics, ESBL and pAmpC β-lactamase producers are also often resistant to quinolones and CT. This is because genes conferring resistance to quinolones and CT have been extensively reported in the same plasmid harboring β-lactamase genes [40]. In addition, ESBL and pAmpC β-lactamase cause multidrug resistance, thus limiting therapeutic choices [41]. There is no resistance to IMP for both ESBL- and pAmpC β-lactamase-producing *E. coli* and *K. pneumoniae*, and the resistance to G was less than 20% for ESBL- and pAmpC β-lactamase-producing *E. coli* and *K. pneumoniae*. Therefore, IMP and G antimicrobials may be potential treatment options for ESBL- and pAmpC β-lactamase-associated infections. The results of the present study could contribute to the improvement of therapeutic guidelines for treating dogs in veterinary hospitals in the USA.

## 4. Conclusions

In summary, we report the development of a method that can detect ESBL- and pAmpC β-lactamase-producing bacteria. This method is a fast, and low-cost tool for the screening of frequently encountered ESBL- and pAmpC β-lactamase-producing bacteria. It would assist in diagnosis and improve therapeutic treatment in animal hospitals. Our study showed that bacteria isolated by this method were identified as ESBL and pAmpC β-lactamase producing bacteria in most cases.

## Figures and Tables

**Figure 1 animals-13-00649-f001:**
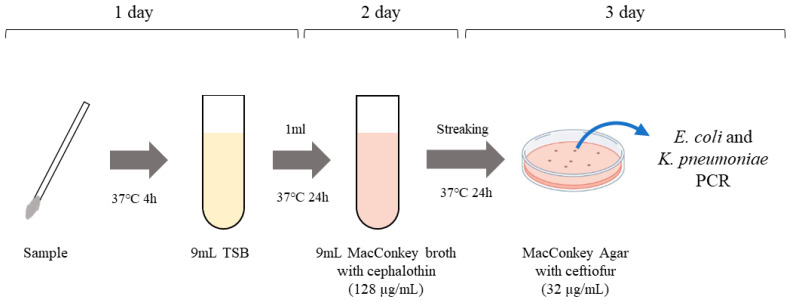
Schematic workflow for isolation of ESBL and pAmpC β-lactamase-producing *E. coli* and *K. pneumoniae*. TSB, tryptic soy broth.

**Figure 2 animals-13-00649-f002:**
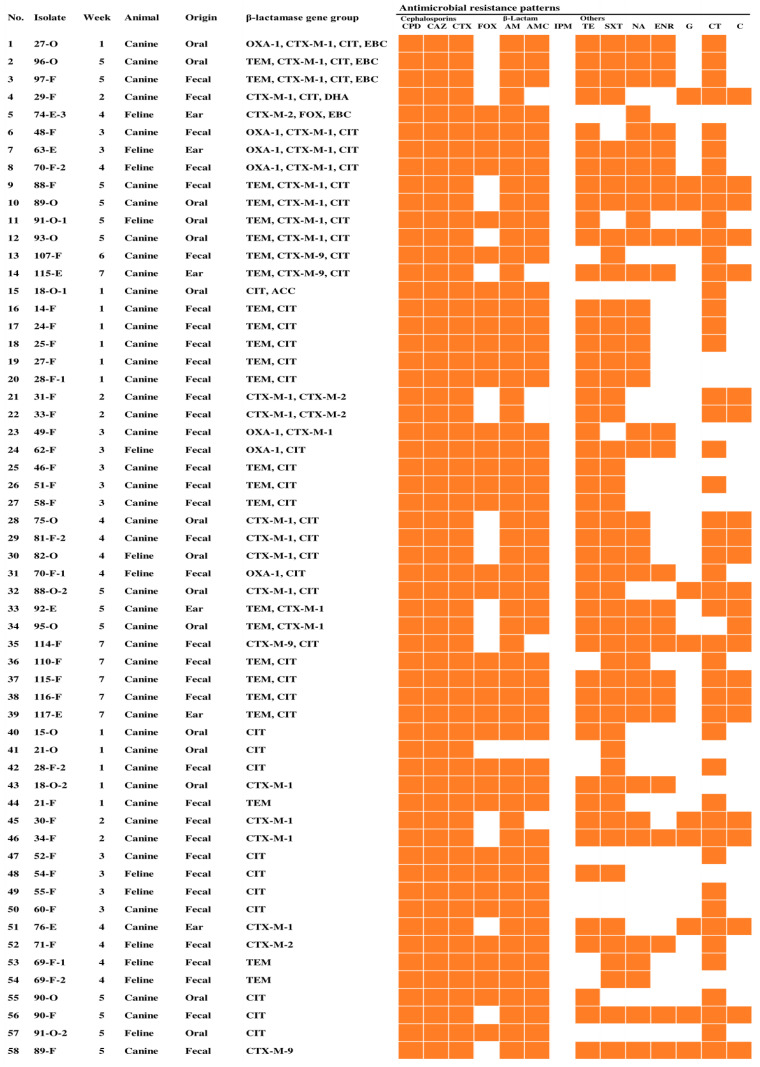
Characteristics of 58 ESBL- and pAmpC β-lactamase-producing *E. coli* isolates from dogs and cats. CPD, cefpodoxime; CAZ, ceftazidime; CTX, cefotaxime; FOX, cefoxitin; AM, ampicillin; AMC, amoxicillin-clavulanate; IPM, imipenem; TE, tetracycline; SXT, trimethoprim/sulfamethoxazole; NA, nalidixic acid; ENR, enrofloxacin; G, gentamicin; CT, colistin; C, chloramphenicol.

**Figure 3 animals-13-00649-f003:**
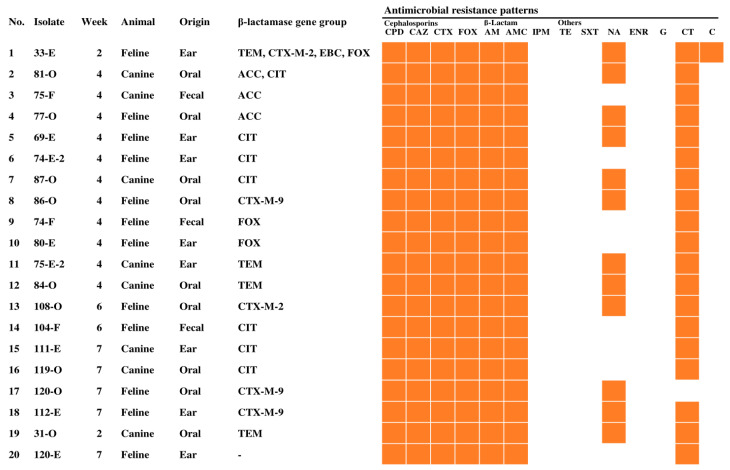
Characteristics of 20 ESBL- and pAmpC β-lactamase-producing *K. pneumoniae* isolates from dogs and cats. CPD, cefpodoxime; CAZ, ceftazidime; CTX, cefotaxime; FOX, cefoxitin; AM, ampicillin; AMC, amoxicillin-clavulanate; IPM, imipenem; TE, tetracycline; SXT, trimethoprim/sulfamethoxazole; NA, nalidixic acid; ENR, enrofloxacin; G, gentamicin; CT, colistin; C, chloramphenicol.

**Table 1 animals-13-00649-t001:** History of *E. coli* and *K. pneumoniae* used in this study for the development of an isolation method.

No.	DDST ^a^ Test	Bacteria	Animal	Origin	ESBL and pAmpC β-Lactamase Genes	Resistance Pattern of Cephalosporins ^b^
1	+	*K. pneumoniae*	Canine	Wound	*TEM*, *OXA-1*, *CTX-M-1*, *CIT*	CEP, FOX, CPD
2	+	*K. pneumoniae*	Canine	Wound	*TEM*, *OXA-1*, *CTX-M-1*, *CIT*	CEP, FOX, CPD
3	+	*K. pneumoniae*	Canine	Wound	*TEM*, *OXA-1*, *CTX-M-1*, *CIT*	CEP, FOX, CPD
4	+	*K. pneumoniae*	Canine	Urine	*TEM*, *OXA-1*, *CTX-M-1*	CEP, FOX, CPD
5	+	*K. pneumoniae*	Canine	Urine	*TEM*, *OXA-1*, *CTX-M-1*	CEP, FOX, CPD
6	+	*K. pneumoniae*	Canine	Abscess	*TEM*, *OXA-1*, *CTX-M-1*	CEP, FOX, CPD
7	+	*K. pneumoniae*	Canine	Wound	*TEM*, *OXA-1*, *CTX-M-1*	CEP, CPD
8	+	*K. pneumoniae*	Canine	Urine	*TEM*, *OXA-1*, *CTX-M-1*	CEP, CPD
9	+	*K. pneumoniae*	Canine	Urine	*TEM*, *OXA-1*, *CTX-M-1*	CEP, CPD
10	+	*K. pneumoniae*	Canine	Urine	*TEM*, *OXA-1*, *CTX-M-1*	CEP, CPD
11	+	*K. pneumoniae*	Canine	Ear	*TEM*, *OXA-1*	CEP, CPD
12	+	*K. pneumoniae*	Canine	Drain	*TEM*, *OXA-1*	CEP, CPD
13	+	*K. pneumoniae*	Canine	Wound	*TEM*, *OXA-1*	CEP, CPD
14	+	*K. pneumoniae*	Canine	Ear	*TEM*, *OXA-1*	CEP, CPD
15	+	*K. pneumoniae*	Feline	Abscess	*TEM*, *OXA-1*	CEP, CPD
16	+	*K. pneumoniae*	Canine	Skin swab	*CIT*	CEP, FOX, CPD
17	+	*K. pneumoniae*	Canine	Wound	*CTX-M-1*	CEP, CPD
18	+	*K. pneumoniae*	Canine	Bronchial aspirate	*TEM*	CEP, CPD
19	+	*K. pneumoniae*	Canine	Biopsy lung	*TEM*	CEP, FOX
20	+	*K. pneumoniae*	Canine	Nasopharyngeal swab	*TEM*	CEP, FOX
21	+	*K. pneumoniae*	Canine	Wound	*TEM*	CEP, CPD
22	+	*K. pneumoniae*	Equine	Catheter	*TEM*	CEP
23	+	*K. pneumoniae*	Equine	wound	*TEM*	CEP
24	-	*K. pneumoniae*	Canine	Wound	-	-
25	-	*K. pneumoniae*	Canine	Urine	-	-
26	-	*K. pneumoniae*	Equine	Gastric aspirate	-	-
27	-	*K. pneumoniae*	Equine	Tracheal aspirate	-	-
28	-	*K. pneumoniae*	Canine	Urine	-	-
29	-	*K. pneumoniae*	Canine	Wound	-	-
30	-	*K. pneumoniae*	Canine	Prostatic fluid	-	-
31	-	*K. pneumoniae*	Canine	Systemic Infection	-	-
32	-	*K. pneumoniae*	Canine	Urine	-	-
33	-	*K. pneumoniae*	Canine	Ear	-	-
34	-	*K. pneumoniae*	Canine	Urine	-	-
35	-	*K. pneumoniae*	Canine	Ear	-	-
36	-	*K. pneumoniae*	Canine	Wound	-	-
37	-	*K. pneumoniae*	Canine	Urine	-	-
38	-	*K. pneumoniae*	Canine	Urine	-	-
39	-	*K. pneumoniae*	Canine	Urine	-	-
40	-	*K. pneumoniae*	Canine	Skin swab	-	-
41	-	*K. pneumoniae*	Canine	Ascitic fluid	-	-
42	-	*K. pneumoniae*	Canine	Urine	-	-
43	-	*K. pneumoniae*	Canine	Urine	-	-
44	-	*K. pneumoniae*	Canine	Urine	-	-
45	-	*K. pneumoniae*	Canine	Urine	-	-
46	-	*K. pneumoniae*	Canine	Urine	-	-
47	+	*E. coli*	Canine	Urine	*TEM*, *OXA-1*, *CTX-M-1*	CEP, CPD
48	+	*E. coli*	Canine	Urine	*TEM, CIT*	CEP, FOX, CPD
49	+	*E. coli*	Canine	Urine	*OXA-1*, *CTX-M-1*	CEP, CPD
50	+	*E. coli*	Canine	Urine	*CIT*	CEP, FOX, CPD
51	+	*E. coli*	Canine	Urine	*CIT*	CEP, FOX, CPD
52	+	*E. coli*	Canine	Urine	*CIT*	CEP, FOX, CPD
53	+	*E. coli*	Canine	Urine	*CIT*	CEP, FOX, CPD
54	+	*E. coli*	Canine	Urine	*CIT*	CEP, CPD
55	+	*E. coli*	Canine	Urine	*CIT*	CEP, CPD
56	+	*E. coli*	Canine	Urine	*CTX-M-1*	CEP, CPD
57	+	*E. coli*	Canine	Urine	*TEM*	CEP
58	+	*E. coli*	Canine	Urine	*TEM*	CEP
59	+	*E. coli*	Canine	Urine	*TEM*	CEP
60	-	*E. coli*	Canine	Urine	-	-
61	-	*E. coli*	Canine	Urine	-	-
62	-	*E. coli*	Canine	Urine	-	-
63	-	*E. coli*	Canine	Urine	-	-
64	-	*E. coli*	Canine	Urine	-	-
65	-	*E. coli*	Canine	Urine	-	-
66	-	*E. coli*	Canine	Urine	-	-
67	-	*E. coli*	Canine	Urine	-	-
68	-	*E. coli*	Canine	Urine	-	-
69	-	*E. coli*	Canine	Urine	-	-
70	-	*E. coli*	Canine	Urine	-	-
71	-	*E. coli*	Canine	Urine	-	-
72	-	*E. coli*	Canine	Urine	-	-

^a^ DDST, double disc synergy test; + indicate that was DDST Test positive; - indicate that was DDST Test negative. ^b^ CEP, cephalothin; FOX, cefoxitin CPD, cefpodoxime.

**Table 2 animals-13-00649-t002:** Prevalence of *E. coli* and *K. pneumoniae* growth under different concentrations of each cephalosporin.

Bacteria	No. of Grown Bacteria (%)						
	First-Generation Cephalosporin	Third-Generation Cephalosporin					
	Cephalothin (μg/mL)	Ceftiofur (μg/mL)			Ceftriaxone (μg/mL)		
	128	16	32	64	16	32	64
ESBL- and pAmpC β-lactamase-producing *E. coli* (n = 13)	13 (100.0)	13 (100.0)	13 (100.0)	7 (53.8)	10 (76.9)	7 (53.8)	5 (38.5)
Non ESBL- and pAmpC β-lactamase-producing *E. coli* (n = 13)	10 (76.9)	5 (38.5)	0 (0.0)	0 (0.0)	3 (23.1)	1 (7.7)	1 (7.7)
ESBL- and pAmpC β-lactamase-producing *K. pneumoniae* (n = 23)	23 (100.0)	23 (100.0)	22 (95.7)	16 (69.6)	20 (87.0)	18 (78.3)	14 (60.9)
Non ESBL- and pAmpC β-lactamase-producing *K. pneumoniae* (n = 23)	17 (73.9)	13 (61.5)	0 (0.0)	0 (0.0)	8 (34.8)	5 (21.7)	4 (17.4)

**Table 3 animals-13-00649-t003:** Distribution of growth under each cephalosporin in samples isolated from dogs and cats.

Total Number of Samples	No. of Samples That Grew or Did Not Grow in MacConkey Broth Containing Cephalothin (%)		No. of Samples That Grew or Did Not Grow on MacConkey Agar Containing Ceftiofur (%)	
109	Grew	99 (90.8)	Grew	71 (65.1)
			Did not grow	28 (25.7)
	Did not grow	10 (9.2)	Grew	0 (0.0)
			Did not grow	10 (9.2)

**Table 4 animals-13-00649-t004:** Distribution of *E. coli* and *K. pneumoniae* isolates from dog and cat samples.

No. of Samples	Bacteria	No. of Bacteria Isolates	No. of Bacteria Isolates Detected β-Lactamase Gene(s)
Grew on MacConkey agar containing ceftiofur			
71	*E. coli*	58	58
	*K. pneumoniae*	20	19
Did not grow on MacConkey agar containing ceftiofur			
38	*E. coli*	32	0
	*K. pneumoniae*	10	0

**Table 5 animals-13-00649-t005:** Distribution of ESBL and pAmpC β-lactamase genes in ESBL- and pAmpC β-lactamase-producing *E. coli* and *K. pneumoniae* isolates from dogs and cats.

Genotype	No. of ESBL and pAmpC β-Lactamase Genes	
	*E. coli*	*K. pneumoniae*
CIT	43	7
TEM	25	4
CTX-M-1	24	0
CTX-M-9	4	3
OXA-1	7	0
CTX-M-2	4	2
EBC	4	1
ACC	1	3
FOX	1	3
DHA	1	0
Total	114	23

## Data Availability

Data is contained within the article.

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
