# Peer review of "Development of a Method for the Fast Detection of Extended-Spectrum β-Lactamase- and Plasmid-Mediated AmpC β-Lactamase-Producing Escherichia coli and Klebsiella pneumoniae from Dogs and Cats in the USA"

_animals, 2023, doi:10.3390/ani13040649_

Round 1
Reviewer 1 Report
Kwang described the establishment of a new method to monitor the fast detection of extended-spectrum β-lactamase- and plasmid-mediated AmpC β-lactamase-producing Escherichia coli and Klebsiella pneumoniae. The overall conclusion is supported by the performed assays. The author is encouraged to address the questions below:
1) The sample collection needs to have IACUC approval, and the consent from the animal owners;
2) The basic methodology needs to be provided briefly in the Abstract of this manuscript. Otherwise, the readers have to search your full manuscript to identify what kind of methods you are establishing in this work.
3) There is a single author for this manuscript. However, in the Author Contributions, you have lots if XX, YY...?
4) The main conclusion of this work is: "This method is a fast, low-cost, and reliable tool 234 for the screening of frequently encountered ESBL- and pAmpC β-lactamase-producing 235 bacteria." I am ok with the part of "fast, low-cost". However, you have to validate the part of this test being "reliable". To validate the test, you have to compare the test established in this work with other gold-standard assays. Then, you can conclude that this test is reliable;
5) The author needs to polish the English writing.
Reviewer 2 Report
The manuscript proposes a laboratory protocol to identify extended-spectrum β-lactamase- and plasmid-mediated AmpC β-lactamase-producing Escherichia coli and Klebsiella pneumoniae.
The work was carried out on a hundred samples and is proposed as a quick and low-cost solution for use in clinical practice.
Congratulations on the work done and on the paper, which is interesting but needs some improvement to give the results the relevance they deserve.
These initial data are interesting, although the method needs to be applied to a larger sample size before it can be considered established. Therefore, I advise the authors to consider and present the paper as a preliminary study and a proposal for the application of a new protocol.
TITLE
Escherichia coli and Klebsiella pneumoniae in italics
AUTHORS
Only one author's name is present, is that correct? Affiliations are numbered but are not present next to the name.
SIMPLE SUMMARY
Line 13 write in full and in italics Escherichia coli and Klebsiella pneumoniae
ABSTRACT
Line 19-20 Antibiotic resistance, such as resistance to beta-lactams and the development of resistance mechanisms, is associated with multifactorial phenomena and not only with the use of third-generation cephalosporins. Correct the sentence
Line 20-21 Escherichia coli and Klebsiella pneumoniae in full and in italics
Line 28 Escherichia coli and Klebsiella pneumoniae in italics
Line 30-32 Escherichia coli and Klebsiella pneumoniae in italics
KEYWORDS
Klebsiella pneumoniae in italics
INTRODUCTION
Line 44 Please add more up-to-date articles:
1. Toombs-Ruane LJ, Marshall JC, Benschop J, Drinković D, Midwinter AC, Biggs PJ, Grange Z, Baker MG, Douwes J, Roberts MG, French NP, Burgess SA. Extended spectrum β-lactamase-and AmpC β-lactamase-producing Enterobacterales associated with urinary tract infections in the New Zealand community: A case-control study. Int J Infect Dis. 2022 Dec 15:S1201-9712(22)00652-X. doi: 10.1016/j.ijid.2022.12.013. Epub ahead of print. PMID: 36529370.
2. Darwich L, Seminati C, Burballa A, Nieto A, Durán I, Tarradas N, Molina-López RA. Antimicrobial susceptibility of bacterial isolates from urinary tract infections in companion animals in Spain. Vet Rec. 2021 May;188(9):e60. doi: 10.1002/vetr.60. Epub 2021 Jan 28. PMID: 33960452.
3. Smoglica C, Evangelisti G, Fani C, Marsilio F, Trotta M, Messina F, Di Francesco CE. Antimicrobial Resistance Profile of Bacterial Isolates from Urinary Tract Infections in Companion Animals in Central Italy. Antibiotics (Basel). 2022 Oct 6;11(10):1363. doi: 10.3390/antibiotics11101363. PMID: 36290021; PMCID: PMC9598067.
4. Yu Z, Wang Y, Chen Y, Huang M, Wang Y, Shen Z, Xia Z, Li G. Antimicrobial resistance of bacterial pathogens isolated from canine urinary tract infections. Vet Microbiol. 2020 Feb;241:108540. doi: 10.1016/j.vetmic.2019.108540. Epub 2019 Nov 29. PMID: 31928695.
I suggest the authors expand the introduction by considering the latest abovementioned data available in the literature.
Furthermore, given the importance of β-lactamase- and plasmid-mediated AmpC β-lactamase-producing E. coli and Klebsiella pneumoniae, I believe it is also relevant to mention the state of the art in considering the spread of these bacteria in the environment and in wildlife. This is important because more and more studies are suggesting a One Health approach to protect humans, animals and the environment.
Please improve the text and add these articles:
6. Solaiman S, Handy E, Brinks T, Goon K, Bollinger C, Sapkota AR, Sharma M, Micallef SA. Extended Spectrum β-Lactamase Activity and Cephalosporin Resistance in Escherichia coli from U.S. Mid-Atlantic Surface and Reclaimed Water. Appl Environ Microbiol. 2022 Aug 9;88(15):e0083722. doi: 10.1128/aem.00837-22. Epub 2022 Jul 12. PMID: 35862684; PMCID: PMC9361821.
7. Chiaverini A, Cornacchia A, Centorotola G, Tieri EE, Sulli N, Del Matto I, Iannitto G, Petrone D, Petrini A, Pomilio F. Phenotypic and Genetic Characterization of Klebsiella pneumoniae Isolates from Wild Animals in Central Italy. Animals (Basel). 2022 May 25;12(11):1347. doi: 10.3390/ani12111347. PMID: 35681810; PMCID: PMC9179660.
8. Palmeira JD, Cunha MV, Carvalho J, Ferreira H, Fonseca C, Torres RT. Emergence and Spread of Cephalosporinases in Wildlife: A Review. Animals (Basel). 2021 Jun 12;11(6):1765. doi: 10.3390/ani11061765. PMID: 34204766; PMCID: PMC8231518.
9. Ballash GA, Dennis PM, Mollenkopf DF, Albers AL, Robison TL, Adams RJ, Li C, Tyson GH, Wittum TE. Colonization of White-Tailed Deer (Odocoileus virginianus) from Urban and Suburban Environments with Cephalosporinase- and Carbapenemase-Producing Enterobacterales. Appl Environ Microbiol. 2022 Jul 12;88(13):e0046522. doi: 10.1128/aem.00465-22. Epub 2022 Jun 23. PMID: 35736227; PMCID: PMC9275232.
10. Darwich, L.; Vidal, A.; Seminati, C.; Albamonte, A.; Casado, A.; López, F.; Molina-López, R.A.; Migura-Garcia, L. High 338 prevalence and diversity of extended-spectrum β-lactamase and emergence of OXA-48 producing Enterobacterales in wildlife 339 in Catalonia. PLoS ONE. 2019, 14, e0210686. (Reference already present in the manuscript)
11. Smoglica C, Vergara A, Angelucci S, Festino AR, Antonucci A, Moschetti L, Farooq M, Marsilio F, Di Francesco CE. Resistance Patterns, mcr-4 and OXA-48 Genes, and Virulence Factors of Escherichia coli from Apennine Chamois Living in Sympatry with Domestic Species, Italy. Animals (Basel). 2022 Jan 6;12(2):129. doi: 10.3390/ani12020129. PMID: 35049753; PMCID: PMC8772577.
Line 66 Please modify the text: “The objective of this study was to develop and evaluate a new detection method”
MATERIALS AND METHODS
Line 128 Why the 2013 guidelines were used?
RESULTS AND DISCUSSION
Line 132 Please write in full Escherichia coli and Klebsiella pneumoniae
Line 139-140 The authors refer to a total of 71 isolates, but 58 Escherichia coli plus 20 Klebsiella pneumonia make a total of 78 isolates. Please, check.
Line 147-148 Is the presence of a new enzyme a result of the study or a hypothesis based on recent data available in the literature? Please clarify.
Line 156 Please write in full Escherichia coli and Klebsiella pneumoniae
Line 178-181 Update the bibliography with the latest manuscripts also including wildlife. As mentioned in the introduction, it is important to emphasise the wide distribution of these bacterial species and their relevance to One Health:
14. Chiaverini A, Cornacchia A, Centorotola G, Tieri EE, Sulli N, Del Matto I, Iannitto G, Petrone D, Petrini A, Pomilio F. Phenotypic and Genetic Characterization of Klebsiella pneumoniae Isolates from Wild Animals in Central Italy. Animals (Basel). 2022 May 25;12(11):1347. doi: 10.3390/ani12111347. PMID: 35681810; PMCID: PMC9179660.
15. Palmeira JD, Cunha MV, Carvalho J, Ferreira H, Fonseca C, Torres RT. Emergence and Spread of Cephalosporinases in Wildlife: A Review. Animals (Basel). 2021 Jun 12;11(6):1765. doi: 10.3390/ani11061765. PMID: 34204766; PMCID: PMC8231518.
16. Yossapol M, Yamamoto M, Sugiyama M, Odoi JO, Omatsu T, Mizutani T, Ohya K, Asai T. Association between the blaCTX-M-14-harboring Escherichia coli Isolated from Weasels and Domestic Animals Reared on a University Campus. Antibiotics (Basel). 2021 Apr 13;10(4):432. doi: 10.3390/antibiotics10040432. PMID: 33924433; PMCID: PMC8069031.
17. Rana C, Rajput S, Behera M, Gautam D, Vikas V, Vats A, Roshan M, Ghorai SM, De S. Global epidemiology of CTX-M-type β-lactam resistance in human and animal. Comp Immunol Microbiol Infect Dis. 2022 Jul;86:101815. doi: 10.1016/j.cimid.2022.101815. Epub 2022 Apr 30. PMID: 35605314.
Line 187 Update the bibliography with the latest manuscripts:
da Silva LCBA, Cardoso B, Fontana H, Esposito F, Cortopassi SRG, Sellera FP, Lincopan N. Human pandemic K27-ST392 CTX-M-15 extended-spectrum β-lactamase-positive Klebsiella pneumoniae: A one health clone threatening companion animals. One Health. 2022 Jul 3;15:100414. doi: 10.1016/j.onehlt.2022.100414. PMID: 36277105; PMCID: PMC9582550.
18. Garcês A, Lopes R, Silva A, Sampaio F, Duque D, Brilhante-Simões P. Bacterial Isolates from Urinary Tract Infection in Dogs and Cats in Portugal, and Their Antibiotic Susceptibility Pattern: A Retrospective Study of 5 Years (2017-2021). Antibiotics (Basel). 2022 Oct 31;11(11):1520. doi: 10.3390/antibiotics11111520. PMID: 36358175; PMCID: PMC9686987.
19. Smoglica C, Evangelisti G, Fani C, Marsilio F, Trotta M, Messina F, Di Francesco CE. Antimicrobial Resistance Profile of Bacterial Isolates from Urinary Tract Infections in Companion Animals in Central Italy. Antibiotics (Basel). 2022 Oct 6;11(10):1363. doi: 10.3390/antibiotics11101363. PMID: 36290021; PMCID: PMC9598067.
20. Darwich L, Seminati C, Burballa A, Nieto A, Durán I, Tarradas N, Molina-López RA. Antimicrobial susceptibility of bacterial isolates from urinary tract infections in companion animals in Spain. Vet Rec. 2021 May;188(9):e60. doi: 10.1002/vetr.60. Epub 2021 Jan 28. PMID: 33960452.
Line 199-220 Authors should decide whether to use the acronym of the antibiotics or the full name and maintain this choice throughout the text. Please check.
CONCLUSION
Line 237-238 Please modify: “Our study showed that bacteria isolated by this method were identified as ESBL and pAmpC β-lactamase producing bacteria in most cases”.
I advise the authors to emphasise that the results are encouraging but need further investigation with more samples.
Round 2
Reviewer 2 Report
Thank you for your contribution
Best regards